# Fibroblasts Influence the Efficacy, Resistance, and Future Use of Vaccines and Immunotherapy in Cancer Treatment

**DOI:** 10.3390/vaccines9060634

**Published:** 2021-06-10

**Authors:** Bailee H. Sliker, Paul M. Campbell

**Affiliations:** 1Cancer Signaling and Epigenetics Program, Fox Chase Cancer Center, Philadelphia, PA 19111, USA; Bailee.Sliker@fccc.edu; 2The Marvin and Concetta Greenberg Pancreatic Cancer Institute, Fox Chase Cancer Center, Philadelphia, PA 19111, USA

**Keywords:** tumor microenvironment, cancer-associated fibroblasts, immunotherapy, cancer vaccines, chemokines, cytokines

## Abstract

Tumors are composed of not only epithelial cells but also many other cell types that contribute to the tumor microenvironment (TME). Within this space, cancer-associated fibroblasts (CAFs) are a prominent cell type, and these cells are connected to an increase in tumor progression as well as alteration of the immune landscape present in and around the tumor. This is accomplished in part by their ability to alter the presence of both innate and adaptive immune cells as well as the release of various chemokines and cytokines, together leading to a more immunosuppressive TME. Furthermore, new research implicates CAFs as players in immunotherapy response in many different tumor types, typically by blunting their efficacy. Fibroblast activation protein (FAP) and transforming growth factor β (TGF-β), two major CAF proteins, are associated with the outcome of different immunotherapies and, additionally, have become new targets themselves for immune-based strategies directed at CAFs. This review will focus on CAFs and how they alter the immune landscape within tumors, how this affects response to current immunotherapy treatments, and how immune-based treatments are currently being harnessed to target the CAF population itself.

## 1. Introduction

Cancer is the second leading cause of deaths in the United States [1]. Despite the high incidence of cancer in the population, it remains difficult to treat for many, and new treatments are being continually developed in the effort to improve survival and quality of life for patients. One development in cancer treatment that has shown promise is the harnessing of the immune system to target the tumor, termed cancer immunotherapy. However, cancer cells do not exist in isolation and are surrounded by a TME that includes multiple other cell types including fibroblasts, immune cells, and endothelial cells [2,3]. This TME plays an ever increasing role in the development of tumors from initiation through to metastasis [4]. Despite the promising results of the different types of cancer immunotherapy, to date, these modalities have not been as effective in certain cancer types as was initially hoped. This is due to a variety of reasons including limited targeting to and accumulation of immune cells within the tumor, the heterogeneous nature of the tumor itself limiting the ability of immune cells to attack the tumor as a whole, and general immune suppression in cancer patients through alterations to chemokine/cytokine secretion as well as subsequent effects on immune cell infiltrates [5,6,7]. One of the major contributing factors to all of these forms of immune suppression observed in cancer patients is the TME. The TME thus plays key roles in both the support and the suppression of many cancers and therefore can also alter the effectiveness of different treatments including immunotherapies [8,9,10]. As fibroblasts are the most abundant non-epithelial cell type within the TME, the goal of this review is to highlight the influence that fibroblasts have on the efficacy of different forms of cancer immunotherapy, including cancer vaccines and immune checkpoint inhibitors. Furthermore, a focus on which proteins contribute to these therapy effects and how they are currently being targeted will also be pursued.

### 1.1. Fibroblasts in Cancer

#### 1.1.1. Normal versus Cancer-Associated Fibroblasts

Fibroblasts makes up a large part of the TME, and not surprisingly, contribute to a myriad of facets of tumor phenotypes and therapy responsiveness. In normal development, the regulation of programs that are restrictive of tumor growth as well as those that promote balance between the growth signals and various assaults to the cell are regulated by fibroblasts [11,12]. Normal fibroblasts will typically undergo activation subsequent to alterations in homeostatic equilibrium such as wound healing, tissue inflammation, and fibrosis. In response to these states, fibroblastic cells, post-activation, build and remodel the extracellular matrix (ECM) [11]. Continued assaults that disrupt this equilibrium, such as those that occur in cancer, ultimately lead to the formation of CAFs and the sustained remodeling of the ECM. CAFs are generally defined as fibroblastic cells that are no longer quiescent and thus considered “active” [13]. They are commonly larger than normal quiescent fibroblasts as well as exhibit increased proliferative and migratory phenotypes. In order to produce ECM as well as growth factors such as chemokines and cytokines, CAFs are also more active metabolically. However, in general, the difference between normal fibroblasts and the different subsets of CAFs is largely functional in nature with the need to compare biomarkers or other detectable features [4]. Certain markers, such as FAPα and alpha smooth muscle actin (αSMA), have been described to distinguish CAFs from the broader fibroblast population [4]. These stromal cells have thus undergone changes, both at the phenotypic and functional level, that allow them to aid in many tumorigenic processes [13]. Specifically, CAFs have been found to play major roles within the TME, where they facilitate the remodeling of the ECM by increased deposition of collagen and fibronectin as well as their cross-linking to promote fibrosis and contribute to tumor progression, metastasis, and drug resistance by releasing soluble factors such as chemokines and cytokines [14,15]. Different CAF populations have also been described [13,16], one of which is the myofibroblastic CAF (myCAF). This type is induced by TGF-β1 or SMAD signaling typically initiated from cytoskeletal alterations, and leads to an ECM that is considered pro-metastatic [17,18]. Conversely, another subtype of CAF includes the inflammatory CAF (iCAF), which has immunosuppressive and immunogenic functions and leads to the secretion of many cytokines and chemokines that are inflammatory in nature, such as inteleukin-6 (IL-6) and chemokine (C-X-C) motif ligand 12 (CXCL12), as well as supportive of tumor progression [16,19]. A third newly discovered subtype of CAFs found through single cell sequencing of mouse and human pancreatic ductal adenocarcinoma (PDAC) tumors is antigen--presenting CAFs (apCAFs) [20]. This population lacks expression of the classic co-stimulatory molecules needed for the induction of T-cell proliferation but does express antigen-presenting molecules including CD74 and major histocompatibility complex (MHC) class II [20]. In turn, antigen presentation to CD4+ T cells by this subtype can occur ex vivo [20]. This function is thus postulated to allow these apCAFs to participate in the immunosuppressive TME of PDAC by inducing anergy of CD4+ T cells upon their binding to the MHC class II on the surface of the apCAF [20]. These different CAF subpopulations are interconvertible despite the fact that they do exhibit distinct molecular signatures [19,21]. Together, these data demonstrate the critical role that CAFs play in a large variety of different cancer-related phenotypes including alteration of the immune landscape within the tumor.

#### 1.1.2. Release of Cytokines and Chemokines

Fibroblasts within the TME are immuno-modulatory (both immunosuppressive and immunogenic in function) with the alterations to production and release of many chemokines and cytokines including IL-8, IL-6, IL-1, IL-2, IL-7, CXCL1, and CXCL12 among many others being one way that they accomplish this [19,20,22,23,24,25]. These chemokines and cytokines have the capacity to affect recruitment of different immune cells, such as cytotoxic CD8+ T cells (CTLs), T helper (Th) cells, and T regulatory cells (Tregs)), tumor-associated macrophages (TAMs), myeloid derived suppressor cells (MDSCs), and tumor-associated neutrophils (TANs) [26]. Furthermore, these factors, either directly or through their immunomodulatory function, can also alter common phenotypes associated with cancer including proliferation, invasion, and metastasis [26,27]. These factors were further implicated in angiogenesis as well as therapy resistance for many different tumor types [26,27]. Because of their integral nature within many diseases, drugs that specifically target critical chemokines and cytokines are in clinical trials for inflammatory-type diseases such as arthritis and asthma. Those that are being tested in cancer, however, focus primarily on CXCR4 antagonists, either through small molecule inhibitors, peptide antagonists, or antibodies [28,29,30,31]. The CXCR2-binding chemokines CXCL1 and CXCL8, as well as CXCR2 itself, have also been a focus for drug development and show promise for the treatment of melanoma, breast cancer, ovarian cancer, and pancreatic cancer in preclinical models [32,33,34,35,36,37]. Inhibitors against many of the other types of chemokines and cytokines in a plethora of different cancer types are also on the rise, several of which are in combination with other forms of immunotherapy or chemotherapy [38]. It is thought that because of their dual effect on both tumor and immune cells, targeting chemokines and their receptors is an ideal form of immunotherapy, but further evaluation on its usefulness in clinical settings is still needed.

### 1.2. Cancer Immunotherapy and Vaccines

#### 1.2.1. Cancer Immunotherapy

Cancer immunotherapy has gained significant traction in the arena of cancer treatments. The field itself was named the Breakthrough of the Year according to *Science* in 2013, and the individuals that discovered the checkpoint molecules and how to inhibit their negative immune regulation were awarded the Nobel Prize in Physiology or Medicine in 2018 [39]. The paradigm of cancer immunotherapy as a whole includes a broad range of different treatment types including checkpoint inhibitors (CPIs), antibodies specific to the tumor, adoptive cell therapy (ACT), oncolytic viruses, and tumor vaccines, all of which are at various pre-clinical and clinical stages [7]. One of the most known and widely used types of cancer immunotherapy are CPIs. These inhibitors are antibodies made against negative regulators of the immune response, which normally function as a break to the immune system, and are upregulated in cancer as a form of immune escape [40]. Programmed death-1 (PD-1) (and its ligand programmed death ligand-1 (PD-L1)) as well as cytotoxic T lymphocyte antigen 4 (CTLA-4) are two common proteins increased on the surface of T cells after T-cell activation, which these antibodies are designed to inhibit. So far, CPIs have been approved for monotherapy use in melanoma, non-small cell lung cancer (NSCLC), head and neck squamous cell carcinoma, renal cell carcinoma, uroepithelial carcinoma, Hodgkin lymphoma, and colorectal cancer with microsatellite instability [41,42,43,44,45,46,47,48,49,50,51,52,53,54].

ACT is another mode of immunotherapy that has shown promise in cancer patients. This process involves infusing T cells (either autologous (from self) or allogenic (from another person)) into the patient. These T cells can be derived from tumor-infiltrating lymphocytes (TILs) or through lymphocytes with T-cell receptors engineered to target a specific cancer antigen (known as chimeric antigen receptor (CAR) T cells) and expanded in vitro prior to reinfusion [55,56,57]. The use of this technique has been attempted in advanced melanoma, and CAR T cells have had clinical success in B cell acute lymphoblastic leukemia and diffuse large B cell lymphoma [57,58,59]. In solid tumors, however, only limited success has been observed to date, but clinical trials in a diverse array of cancer types using multiple CAR T cells are currently underway [60,61,62].

An added form of immunotherapy also involves the use of oncolytic viruses. These viruses, which include a wide range such as adenovirus, herpesvirus, and cocksackievirus among many others, are able to selectively target cancer cells through multiple mechanisms including highly expressed viral entry receptors on tumor cells, rapid cell division supporting viral replication, and tumors exhibiting antiviral type I interferon signaling deficiencies, and activate a potent anti-tumor immune response through the release different tumor-specific antigens that can be recognized by antigen-presenting cells (APCs) and adaptive immune cells [63,64,65,66]. There has been one oncolytic virus (talimogene laherparepvec, also known as T-VEC) approved by the U.S. Food and Drug Administration (FDA) for use in patients with melanoma lesions that are injectable but non-resectable in the skin and lymph node [67]. This virus is derived from the herpes simplex virus 1, and in clinical trials (NCT00769704), patients treated with this therapy had a significant increase in durable response rate, overall response rate and overall survival compared to granulocyte-macrophage colony-stimulating factor (GM-CSF) alone with minimal severe side effects [68,69,70]. Overall, however, the use of these viruses is still in its infancy in terms of cancer treatment strategies [67].

#### 1.2.2. Cancer Vaccines

Use of vaccines in the context of cancer falls into two categories. They can be prophylactic, similar to the use of many of the widely known vaccines, or in contrast, their implementation can be therapeutic, being used to treat an already established tumor. Those that are prophylactic include agents against human papilloma virus and hepatitis B used in the prevention of cervical cancer and hepatocellular carcinoma, respectively [71]. Those that are therapeutic in nature, however, have the main goal of inducing an adaptive immune response against a patient’s tumor that is already present. One initial example is the use of Bacillus Calmette-Guérin (also known as BCG), an attenuated strain of *Mycobacterium bovis* that has been approved for use since 1990 in bladder carcinoma [72]. Vaccines against cancer have continued to gain traction as a form of therapy due to their ability to induce tumor-specific cytotoxic CD8+ T cells, which are necessary for tumor regression. Early approaches to cancer vaccines involved the use of autologous vaccines where tumor cells derived from the patient were combined with virus or adjuvant to stimulate a polyclonal immune response [73]. One of the most notable versions of cell-based cancer vaccines, however, are called dendritic cell (DC) vaccines. Dendritic cells have the typical function of presenting antigens to T cells, in turn leading to T-cell activation. To create the most common type of DC vaccine, DCs are cultured (initially derived from CD14+ monocytes cultured with GM-CSF and IL-4) with the desired antigen ex vivo and injected back into the patient from which they originated [74]. Sipuleucel-T falls under this category and is one of the only vaccines approved so far by the FDA for use in advanced hormone-therapy resistant prostate cancer [75]. Low toxicity is typically associated with these agents, and they have been tested in clinical trials for a multitude of cancers such as melanoma, acute myeloid leukemia, myeloma, head and neck cancers, and ovarian carcinoma, but low efficacy has been observed [76,77,78,79,80,81]. The use of personalized recombinant vaccines to target tumors utilizing genomic DNA sequenced from the patient is also under development. DNA vaccines are the most notable type within this category and are in Phase I and Phase II clinical trials for various cancer types including breast, prostate, cervical, and ovarian, among many others, but none to date have received FDA approval [82]. Despite the promise of cancer vaccines as effective therapeutic modalities, many demonstrate limitations in their use including difficulties in obtaining cells from the tumor to create the vaccine, the immunogenicity of the neoantigens, and the obstacle of the limited ability of the DC-activated T cells to enter the tumor [83]. The latter has been found to be largely mediated by the many modes of immunosuppression enacted by the TME. Thus, modulation of the TME is a potential way to increase the efficacy of many different immunotherapies, including cancer vaccines.

## 2. Effects of Fibroblasts on Response to Cancer Immunotherapy Treatment

### 2.1. Fibroblasts and Their Effects on Immune Cells and Immunotherapy Treatment

One of the major contributions of CAFs to the TME is the phenotypes of immunosuppression [84]. Typically, this involves exclusion of anti-tumor immune cells, such as CTLs and natural killer (NK) cells, from the tumor space and recruitment of immune suppressing cells, such as Tregs, MDSCs, TAMs, and TANs [84]. Further discussion on the effects of CAFs on the immune cells of the innate and adaptive systems is explored below.

#### 2.1.1. Fibroblasts and Innate Immune Cells

TAMs are one of the most discussed cell types that are affected by CAFs in the stromal compartment [85]. The two distinct subtypes, classified as M1 or M2, are considered anti-tumor and pro-tumor, respectively [85]. M2 macrophages are most relevant in the tumor space, where they secrete factors that are tumor-promoting while also working to suppress immune function by inhibiting the activity of CTLs, and they are able to further inhibit T-cell activation by expressing CPI target proteins [85]. Typically, CAFs recruit monocytes to the TME, where they support differentiation of these monocytes into M2 macrophages [86]. This recruitment is promoted by CAF secretion of IL-6 and monocyte chemoattractant protein 1. The cooperation between these two cell types is also relevant to disease progression, as they together support cancer cell invasion [87]. Furthermore, co-expression of M2 macrophages and αSMA/FAP positive CAFs is associated with poor prognosis in colorectal cancer and oral squamous cell carcinoma patients [88,89], again indicating that improvements in therapy require addressing the functionality of and contribution from a multiplicity of TME cohorts while concurrently targeting epithelial tumor cells.

In addition to TAMs, TANs are also immunosuppressive cells that exist within the TME and are newly considered to play a role in tumor immune responses. It is thought that CAFs aid in the recruitment of these neutrophils through known chemokines and cytokines such as CXCL1, CXCL2, CXCL8, and CCL2 [90]. Like TAMs, TANs are also linked to poor prognosis in patients with renal, pancreatic, colon, melanoma, and head and neck cancers [90]. TANs further serve to promote tumor progression while inhibiting CD8+ T cells similar to that observed with TAMs [91,92].

Within the TME, NK cells are a further chief innate immune cell type affected by CAFs [93]. NK cells are cytotoxic and they produce various chemokines and cytokines that lead to maturation of APCs. They are typically influenced by CAFs through CAF secretion of TGF-β, which inhibits NK cell activation and cytotoxic activity through reduction of their activation receptors and a decrease in their cytotoxic enzymes granzyme B and perforin [94,95,96,97]. Similar to the other innate cell types discussed, NK cells also affect the survival of PDAC patients as the number of NK cells within the tumor is correlated with both reduced stroma and a subsequent increase in survival of these patients [98].

#### 2.1.2. Fibroblasts and Adaptive Immune Cells

In addition to cells of the innate immune system, CAFs also affect cells of the adaptive immune system. One of these cell types and the most critical to anti-tumor immunity is T cells. CAFs play a major role in the control of T-cell function and activity within the tumor space. TGF-β plays a sizable part in this regulation, where it promotes death of cytotoxic CD8+ T cells through survival protein inhibition, as well as blocking the expression of perforin and granzymes, both of which are needed for their cytotoxic activity [99,100,101]. CXCL12 and other factors produced by CAFs also reduce T-cell movement and recruitment into the tumor [102]. Additionally, CAFs stimulate the differentiation of CD4+ helper T cells into more immunosuppressive subtypes such as Tregs (Th17) as well as Th2 (tumor-promoting) T cells [103,104,105,106,107]. Furthermore, the ECM made by CAFs is considered to be a physical barrier to T-cell infiltration into the tumor, thereby preventing their interaction with tumor cells themselves, as seen in PDAC and lung cancer [108,109,110,111].

In addition to direct effects on T cells, CAFs can also affect other innate immune cells that, in turn, have the capability to further alter the anti-tumor effects of T cells. Dendritic cells are one such cell type, and they are critical APCs within the TME, as they aid in T-cell activation [112]. One way that these cells are influenced by CAFs occurs through CAF secretion of TGF-β, which leads to a reduction in the co-stimulatory molecules on the surface of DCs, thereby altering their ability to present antigens [113,114,115]. This, in turn, can affect the recruitment of T cells and CTL activation, and can lead to formation of Tregs, whose main role is to inhibit the function of other anti-tumor T cells [114,115]. In addition to TGF-β, vascular endothelial growth factor (VEGF) secretion by CAFs can also block generation and maturation of DCs, again reducing their crucial ability to present antigens [116,117,118,119].

Finally, MDSCs represent another type of immune cell modulated by CAFs in the stromal compartment of the tumor that can disrupt the function of T cells as well as many other immune cells subtypes [120,121]. These cells are a heterogeneous population of myeloid cells that are immunosuppressive in nature [120,121]. They typically secrete many factors that alter the function of T cells as well as DCs, NK cells, and macrophages [122,123]. In many cancers, including pancreatic cancer, CAFs support monocytes differentiating into MDSCs through IL-6 secretion and STAT3 signaling, modulating T-cell function and proliferation [122,123]. The presence of these cells can also alter the survival of patients, with higher numbers correlating with poorer survival [124].

#### 2.1.3. Fibroblasts and Efficacy of Immunotherapy

More evidence is emerging that response to different immunotherapy approaches is due to more than just the immunogenicity of the cancer cells themselves. A prominent reason that these cancer immunotherapies have not been as promising in some tumors as in preclinical models is the immunosuppressive TME [22,125]. There are three main ways that fibroblasts can contribute to this (Figure 1). The first is the physical barrier that CAFs create through ECM deposition that leads to the exclusion of any T cells that could have been activated through immunotherapy [126]. This has been shown in models of pancreatic cancer, and this ECM deregulation is one of the biggest indicators of poor immunotherapy response, especially with CPIs [127]. Secondly, fibroblasts are able to increase production and secretion of immunosuppressive cytokines. These cytokines, as covered in an earlier section, alter both immune cell recruitment and function. Furthermore, they have been found to increase the expression of checkpoint molecules on cancer cells through CXCL5, CXCL1, and CXCL2 chemokine signaling in addition to fibroblasts also expressing these molecules, with this effect being observed in colorectal cancer and melanoma [128,129,130,131,132]. The final way is to abrogate and functionally affect the immune cells that do infiltrate, as reviewed above. This generally leads to the presence of cell types that promote immunosuppression. CAFs, thus, play major roles in both tumor progression and immune modulation in a variety of different ways and contribute to the reduction in efficacy of many different cancer-targeting therapies including immunotherapy.

### 2.2. Predictors of Immunotherapy Response and Attempts at Their Therapeutic Targeting

In response to the effects of fibroblasts on the efficacy of immunotherapy, studies have been conducted to investigate critical CAF proteins and how they are related to immune cells and response to immunotherapies. Two of the best studied thus far are FAP and TGF-β. The relation of each of these proteins to immunotherapy response and current therapeutic strategies to mitigate these effects are discussed below.

#### 2.2.1. Fibroblast Activation Protein

FAP, a transmembrane serine protease and marker of CAF activation, is exclusively synthesized and highly expressed by CAFs of nearly all epithelial cancers while being relatively absent on normal stromal cells [133,134,135,136,137]. Additionally, it plays a large role in tumor progression and metastasis, and its presence in histological staining is correlated to poor prognosis for most cancer patients [138,139]. In regards to immunotherapy, previous studies have shown that when FAP is genetically depleted in PDAC and lung carcinoma models, this led to hypoxic necrosis in the tumor in addition to an induction of CD8+ T-cell infiltration [140]. Furthermore, in a study of 117 melanoma patients treated with anti-PD-1 therapy, a positive association between progression free survival and overall survival was associated with high FAP cell count. In comparison, the opposite was true in patients not treated with immunotherapy, where FAP became a significant negative prognostic biomarker [141]. This study thus classified FAP as a negative prognostic but positive indicative biomarker in melanoma and postulated that FAP may have a role in survival advantage after PD-1 targeted therapy [141]. In response to this crucial role of FAP in tumor progression, anti-FAP therapies have been developed in an effort to target CAFs. One way that this has been done is through a small molecule inhibitor of the peptidyl peptidase activity of FAP [107]. This small molecule, called PT-630, inhibits the cell surface dipetidyl peptidase activity of FAP. PT-630 has been used in breast, colon, lung, PDAC, and oral squamous cell carcinoma models, where it showed an ability to induce apoptosis and CTL infiltration as well as reduce cell proliferation and metastasis [142,143,144,145]. Furthermore, PT-100 is another inhibitor of FAP that works in a similar mechanism to PT-630, but it also inhibits its intracellular activity. In mice, PT-100 led to a reduction in tumor growth in syngeneic mouse models of fibrosarcoma, lymphoma, melanoma, and mastocytoma [146]. Additional experiments revealed not only a reduction in tumor growth but also rejection of tumors mediated through CTLs that were tumor specific as well as the occurrence of immunologic memory [146]. Likewise, upregulation of chemokines and cytokines known to attract and prime T cells was also observed upon treatment with PT-100 [146]. Another inhibitor of FAP includes promelittin protoxin, an engineered form of the toxic component in European honeybee venom that can now be cleaved by FAP, in turn making FAP-positive cells sensitive to its toxic effects. The use of this protoxin was found to selectively kill cell lines expressing FAP, and in xenografts of human breast and prostate cancer, significant cell lysis and subsequent growth inhibition were observed with little toxicity in host mice [147]. Additionally, the use of monoclonal antibodies against FAP is another mode of therapy targeting CAFs. These have been used in a clinical study of patients with colorectal carcinoma or NSCLC. Despite these antibodies showing preferential uptake by tumor tissue, significant clinical responses were not observed in a majority of patients [148]. One such antibody, sibrotuzumab, showed some feasibility with low toxicities in Phase I clinical trials of colon cancer but showed low efficacy in Phase II trials [148,149,150].

#### 2.2.2. TGF-β

The ECM has been found to be a critical player in many different cancer types, where it regulates processes such as cancer cell growth and metastasis [151]. It can further be used to predict prognosis in some cancers such as pancreatic and colorectal [152]. TGF-β in the TME has been implicated as a potent regulator of the ECM. Its secretion by CAFs occurs upon their activation and, through an autocrine loop, sustains their differentiation. One study showed that, in a large pan-cancer analysis, an ECM transcriptional program upregulated in cancer is correlated with the activation of TGF-β signaling from CAFs [153]. Furthermore, this was linked to immunosuppression in tumors and was a predictor for failure of PD-1 inhibitors [153]. It was postulated that transcriptome normalization in CAFs, through something like TGF-β inhibition, in combination with PD-1 inhibitors, could overcome these effects [153]. This has been demonstrated in a few different mouse models where concurrent inhibition of TGF-β and PD-1 through a dual targeting antibody specific to both PD-L1 and TGF-βRII contributed to reduced tumor growth and metastasis [154]. Additionally, this bi-efficacious antibody showed greater response in mouse models than single therapy alone and conferred long term anti-tumor immunity [154]. Furthermore, in a murine model of colon cancer, when TGF-β was inhibited, an increase in cytotoxic T-cell tumor-specific response was observed, and liver metastatic disease became more susceptible to anti-PD-1-PD-L1 therapies [155]. This study suggested that TGF-β present in the TME contributed to a mechanism of immune evasion within these tumors [155]. A recent study by Kieffer et al. found by single cell RNA sequencing analysis of CAFs from breast cancer patients that, among the five most abundant clusters of CAF subtypes, the TGF-β-myCAF cluster was associated with an immunosuppressive environment [156]. This cluster was correlated with the amount of PD-1+ and/or CTLA4+ CD4+ T cells as well as negatively correlated with CD8+ T-cell infiltration [156]. Furthermore, this cluster was enriched at the time of diagnosis in patients with melanoma or NSCLC who did not respond to immunotherapies [156]. Additionally, TGF-β1 and - β3 are highly expressed by TGF-β-myCAFs, which is critical, as TGF-β has been found to abrogate response to anti-PD-L1 by excluding T cells from the tumor, and its inhibition led to a potent T-cell response against tumor cells [156]. Separately, in a model of melanoma, the use of pharmacological inhibitors of the TGF-β signaling pathway was synergistic with CTLA4 inhibitors [157]. Also in this study, TGF-β inhibition led to cleavage of PD-L1 through a matrix metallopeptidase 9 (MMP-9)-dependent mechanism contributing to resistance to PD-1 therapies [157]. This study also indicated that MMP-9 can desensitize tumors to therapies against PD-1 [157]. Finally, TGF-β inhibitors have also been combined with dendritic cell-based vaccines in models of lymphoma, lung cancer, and melanoma, and synergy was found in all models [158].

## 3. Targeting of CAFs with Immunotherapy

Due to the ever-growing nature of the effects of CAFs on tumor progression, immunotherapy technologies are currently under development to target CAF populations themselves in an effort to halt disease advancement. Two main categories currently exist for these therapy types and include vaccines and CAR T-cell therapy, both of which will be explored in detail below.

### 3.1. CAF-Targeting Vaccines

In addition to the aforementioned therapies against FAP, the use of vaccines against this protein has also been explored (Figure 2) [107,142,159,160]. Early studies that were conducted with dendritic cells transfected with mRNA that encodes FAP led to a reduction in tumor growth in various different tumor types [161]. Additionally, a subsequent study from this same group using mouse models of melanoma and lymphoma showed that vaccination of mice with DCs loaded with FAP mRNA reduced primary tumor growth as well as lung metastases [159]. In models of multidrug resistant murine colon and breast cancers, the use of a FAP DNA vaccine taken orally in a prophylactic setting led to a decrease in tumor growth [160]. When used in a therapeutic setting, a similar response was observed in the growth of both primary tumor and lung metastases through a CD8+ T-cell-mediated mechanism [160]. Further data indicated that use of this vaccine reduced collagen type I and subsequently increased chemotherapy uptake [160]. Thus, the use of this vaccine along with common chemotherapy drugs such as doxorubicin enhanced the effects of these agents as compared to single use alone and led to not only a decrease in primary tumor growth but also a 50% complete rejection of tumors in mouse models and a three-fold increase in survival [160]. In later studies by this same group, further analysis on the immune effects of this FAP-targeting DNA vaccine found a shift in the immune environment towards the more anti-tumor Th1 polarization with an increase in expression in the Th1 cytokine profile including IL-2 and IL-7 [107]. Decreases in known immunosuppressive cells including TAMs, MDSCs, and Tregs as well as an increase in DCs and cytotoxic CD8+ T cells were also observed [107]. Furthermore, in this model, the use of the FAP vaccine led to a reduction in tumor angiogenesis and lymphangiogenesis [107]. Similarly, another DNA vaccine against FAP constructed by Wen and colleagues also showed analogous results where its administration led to reduction in tumor growth and increased survival in tumor-bearing mice in a model of colon cancer [142]. CTL activation was again found to be critical with this vaccine against FAP and is thought to be the mechanism through which the inhibition of tumor growth is mediated [142]. An additional study using a replication-deficient adenovirus-based vaccine that expresses FAP showed that, in a model of mouse melanoma, combination treatment with this agent and a traditional vaccine against melanoma-associated antigens enhanced the effects of the traditional vaccine against tumors [162]. Furthermore, increases in CTL frequency and effector function were observed with concomitant decreases in their exhaustion. The use of this therapeutic also diminished immunosuppressive cells through reduction of chemokines and cytokines C-C chemokine ligand 5 (CCL5), CCL22, IL-4, IL-10, and TGF-β [162]. Moreover, a further study used a DNA vaccine that targeted both FAP in CAFs and survivin, a critical inhibitor of apoptosis, in tumor cells. This vaccine, termed OsFS, showed anti-tumor efficacy in a model of breast cancer [163]. When combined with doxorubicin pretreatment to remove peripheral MDSCs that typically contribute to immunosuppression, the anti-tumor activity of OsFS was amplified [163]. Together, this study showed that inhibiting both CAFs as well as immune cells typically present in the TME, such as MDSCs, contributes to an increase in functional tumor-infiltrating lymphocytes and better anti-tumor activity of these treatments. A reduction in spontaneous lung metastases of this model was also found to be reduced when combination therapy was used [163].

Similar strategies have also been implemented to target TGF-β in the TME. One such agent is a TGF-β2 antisense vaccine. Used in preclinical studies of glioma, it showed improved survival in mice [164]. Furthermore, in a Phase II study of NSCLC (NCT01058785), a different TGF-β2 antisense inoculation, belagenpneumatucel-L, showed similar results [165,166]. A Phase III trial (NCT00676507) investigated the effects of this therapeutic on overall survival of NSCLC patients post-chemotherapy, but unfortunately significant increases in survival were not observed [167].

### 3.2. CAR T-Cells

CAR T-cell therapy has mainly focused on targeting cancer cells themselves, but recently the focus has shifted to targeting parts of the stroma. Non-cancer cells are an attractive target for CAR T cells as compared to cancer cells for a variety of reasons. Some of these include the fact that stromal cells are genetically more stable, their significant role in tumor progression and resistance to therapy, and that therapies targeting them can be used against a variety of different cancers because the mechanisms by which the stroma supports cancer cells are shared among many different tumor types [168]. One such innovation has been the development of CAR T cells targeting FAP using a single-chain Fv FAP monoclonal antibody [169]. These T cells secreted interferon gamma (IFNγ) as well as were targeted in their killing of FAP-positive 3T3 target cells [169]. When adoptively transferred into mice, there was a reduction in highly expressing FAP stromal cells as well as a reduction in the tumor growth of subcutaneous tumors transplanted into mice [169]. The use of these CAR T cells in combination with a tumor vaccine (Ad.E7 vaccine) led to an even greater anti-tumor response as compared to each individual treatment alone [169]. Additionally, minimal side effects were observed, suggesting the movement forward toward clinical development [169]. Three studies published around the same time also investigated the use of CAR T cells targeting FAP-expressing cells with various and sometimes conflicting results. One study that combined anti-FAP CAR T cells along with total body irradiation and IL-2 injections showed limited anti-tumor efficacy as well as enhanced side effects including bone marrow toxicity and cachexia in numerous tumor types such as those of the head and neck, lung, and pancreas [143,170]. Another study showed anti-tumor efficacy and no side effects in an immunodeficient mouse model of lung cancer, while the final study, done in a mesothelioma model, showed survival benefits for tumor-bearing mice, but toxicities could not be evaluated, as the antibody did not react with mouse FAP [137,171,172,173]. In PDAC, known to be an immunologically “cold” tumor, the use of FAP-specific CAR T cells restrained tumor growth as well as reduced vasculature in the tumor through a mechanism that was immune-independent but stromal-dependent [174].

Direct kinase inhibitors of TGF-β exhibit safety concerns due to the cytokine’s control of tissue remodeling and immunomodulation; therefore, the adoptive cell transfer of T cells resistant to TGF-β’s immunosuppressive effects is promising [175,176]. These engineered cells express a dominant negative form of the TGF-β receptor [177]. In models of prostate cancer where CAR T cells directed against prostate-specific membrane antigen also co-expressed a dominant-negative form of TGF-βRII [178], there was an increase in the proliferation of lymphocytes and cytokine secretion, and the CAR T cells also exhibited resistance to exhaustion and persistence in vivo [178]. Furthermore, a near complete loss of tumors in mouse models was observed upon treatment [178]. These promising results have led to a currently active Phase I clinical trial for use in prostate cancer (NCT03089203). In another Phase I study of patients with refractory Epstein Barr virus-positive Hodgkin lymphoma (NCT00368082), the use of engineered T cells with dominant negative TGF-βRII resulted in robust anti-tumor activity, with two patients showing complete response [179]. Overall, the use of both CAR T cells and TILs expanded ex vivo have been utilized in the targeting of CAFs and have indicated altered TMEs and increased immune responses in humans with cancer.

## 4. Conclusions and Future Directions

Numerous studies in recent years have uncovered the critical roles that CAFs play not only in tumor progression but also in altering immune cell function both through direct effects on immune cell types as well as through the secretion of critical chemokines and cytokines. These investigations have also provided valuable insight into how CAFs and the surrounding stroma can affect the efficacy of different therapeutic interventions including immunotherapy. CAF markers such as FAP and TGF-β have been shown to correlate with response to immunotherapy and have been a major focus of targeting CAFs with both small molecule inhibitors as well as with newly developed immunotherapies. As studies continue to evaluate the role that CAFs and the TME have on both immune cells and on the efficiency of immunotherapies in cancer, further targets are likely to be identified, and novel compounds can be developed to target them. This will also lead to improvement in the efficacy of currently available immunotherapies such as CPIs and cancer vaccines in numerous cancer types. Ultimately, targeting CAFs presents as therapeutically valuable due to the crucial role they play in supporting many tumor types, and the development of immunotherapies that work to eliminate or suppress their immunogenic function opens up potential treatments in a large quantity of patients. If the use of vaccines and CAR T cells against CAF populations continues to show promise in clinical trials, application of these in conjunction with other currently available inhibitors presents an avenue of therapeutic intervention that could affect the disease outcome in many patient populations.

## Figures and Tables

**Figure 1 vaccines-09-00634-f001:**
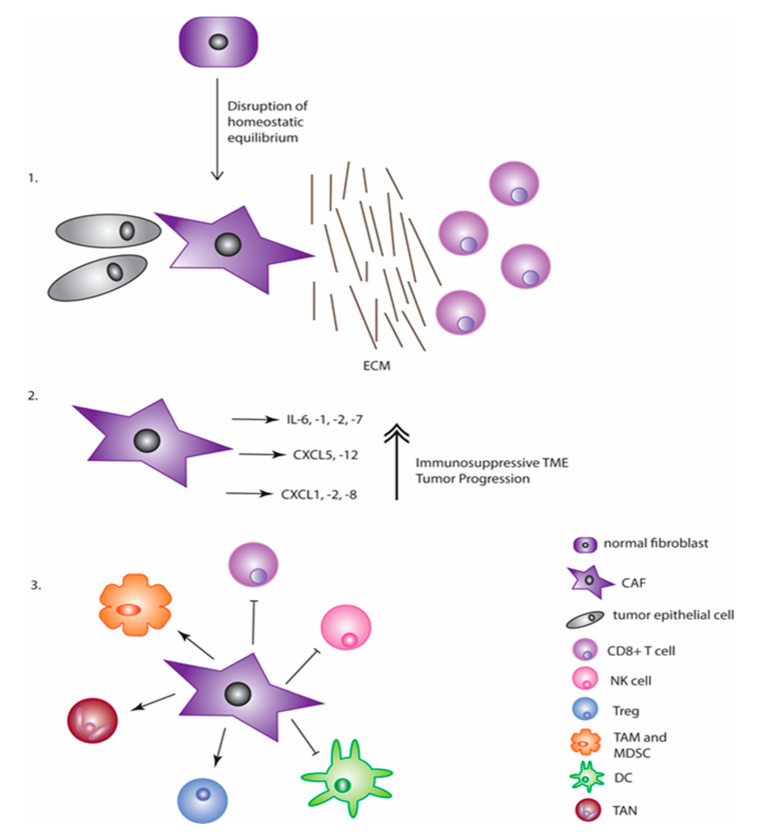
CAFs contribute to an immunosuppressive TME in three major ways. The first is through secretion of ECM proteins that physically separate immune cells (like the CD8+ cytotoxic T cells pictured) from the actual tumor cells. The second is through secretion of critical chemokines and cytokines that aid in both tumor promotion and immune suppression. The third and final way is through suppression of anti-tumor immune cells and promotion of immune cells into the tumor space that dampen the immune response.

**Figure 2 vaccines-09-00634-f002:**
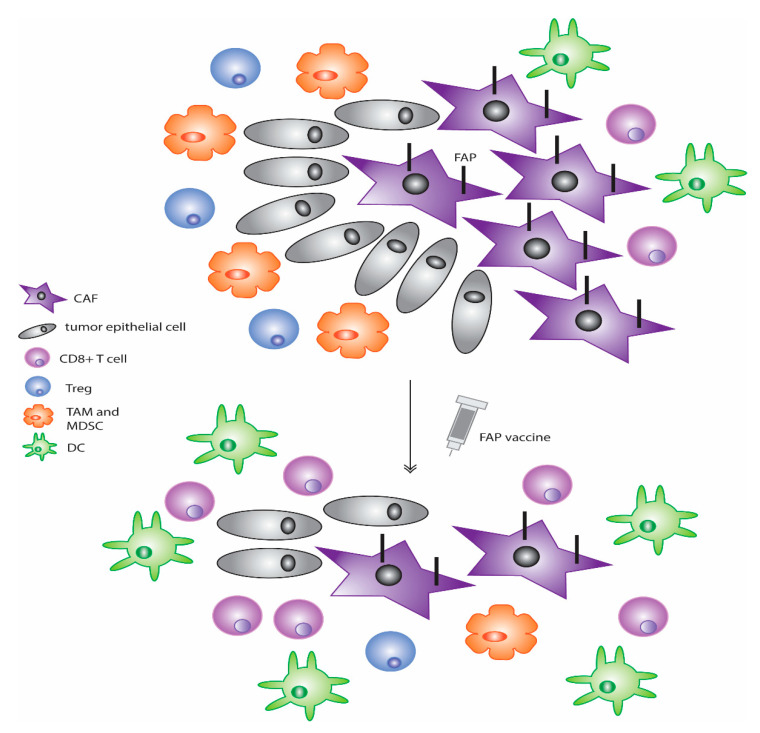
Vaccines against FAP are a new way to target the CAF population in many tumor types. These agents have been shown in multiple cancer models to decrease primary tumor as well as metastatic growth, increase CD8+ T-cell and DC activity, decrease CD8+ T-cell exhaustion, and lessen immunosuppressive cells such as Tregs, TAMs, and MDSCs, as well as reduce the number of FAP-positive CAFs present in the tumor.

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
