# Peer review of "Fibroblasts Influence the Efficacy, Resistance, and Future Use of Vaccines and Immunotherapy in Cancer Treatment"

_vaccines, 2021, doi:10.3390/vaccines9060634_

Round 1
Reviewer 1 Report
In this manuscript, the authors provide in-depth review of the role of cancer-associated fibroblasts in tumor microenvironment, and their influence on the immune responses within the tumors as well as the effect on the efficacy of cancer immune therapy.
This review is very well written It explores the topic of great interest as different modes of therapy targeting cancer-associated fibroblasts are gaining traction. I would recommend the publication of this review with a minor change in the layout of the manuscript as suggested below.
Please move the section 1.1. Cancer Immunotherapy and Vaccines (including section 1.1.1. and section 1.1.2.) to later part of the review. And move the section 1.2. Fibroblasts in Cancer to right after the Introduction.
Author Response
We would like to thank the reviewer for his/her insight. In response, we have reorganized the layout and order of, and updated, Sections 1.1 and 1.2 as per the suggestions.
We have edited and polished the wording, grammar, and punctuation throughout the document.
Reviewer 2 Report
The authors have submitted a comprehensive review on an important issue not been properly considered as much as it is worthy to be
CAF and, more in general TME, are likely to play a paramount role in selecting patients candidate to immunotherapy and/or improve the response to the different immune-therapeutic strategies
The Review is well written and clear. I have no major concerns
Author Response
The authors would like to thank the reviewer for her/his insight and suggestions. In response, we have edited the language, grammar, spelling and punctuation throughout the revised manuscript (these can be seen in alternate colors with the Track Changes feature).
We trust that the revisions and adjustments meet with the reviewer's approval.
Reviewer 3 Report
Sliker and Campbell composed a sound review about the multifaced role of CAFs within the tumor microenvironment with a particular focus on their role in affecting immune cell activity and function and subsequently conditioning the outcome of immunotherapies. They also describe which strategies, including immune-based ones, are currently being investigated to target this population. However, the review still needs some small adjustments:
- Title should be rewritten to better sound
- Within the Abstract (page 1 line 16): Furthermore, new research implicates CAFs as mediators of immunotherapy response in many different tumor types typically by blunting their efficacy. I would suggest to change the word mediators with players
- Page 5 line 223: TAMs are one of the most discussed cell types that CAFs affect in the stromal compartment: affected by CAFs
- Please re-write paragraph 2.1.2 to only include cells belonging to the adaptive immune system. Or better specify the indirect effects that DCs and MDSCs have on the adaptive arms of the anti-tumor response. Please start the paragraph on the T cells then move on describing how these cells could become indirectly affected by the activity exerted by CAFs on innate immune cells such as DCs and MDSCs through their major role in tuning the anti-tumor immune response.
- Minor punctuation and English editing are recommended
Author Response
We thank the reviewer for his/her detailed critiques and suggestions. In response, we have
Adjusted the wording of the title.
Edited the abstract as suggested.
Corrected the text on Page 5.
We have edited, rewritten, and augmented Section 2.1.2 to begin with CAF effects on T cells, and then extended to that of dendritic cells and MDSCs.
We have edited the entire manuscript for language, punctuation, grammar, and style.
We hope that the improved submission meets with the approval of the reviewer and the editors.